# Evidence for adolescent length growth spurts in bonobos and other primates highlights the importance of scaling laws

Andreas Berghaenel[1]*[†], Jeroen MG Stevens[2,3,4][†], Gottfried Hohmann[5,6], Tobias Deschner[5,7], Verena Behringer[5,8,9]

[1]Domestication Lab, Konrad Lorenz Institute of Ethology, Department of Interdisciplinary Life Sciences, University of Veterinary Medicine Vienna, Vienna, Austria; [2]Behavioral Ecology and Ecophysiology, Department of Biology, University of Antwerp, Antwerp, Belgium; [3]Centre for Research and Conservation, Royal Zoological Society of Antwerp, Antwerp, Belgium; [4]SALTO Agro- and Biotechnology, Odisee University of Applied Sciences, Sint-Niklaas, Belgium; [5]Max Planck Institute for Evolutionary Anthropology, Leipzig, Germany; [6]Max-Planck-Institute of Animal Behaviour, Radolfzell, Germany; [7]Comparative BioCognition, Institute of Cognitive Science, University of Osnabrück, Osnabrück, Germany; [8]Endocrinology Laboratory, German Primate Center, Leibniz Institute for Primate Research, Göttingen, Germany; [9]Leibniz ScienceCampus Primate Cognition, German Primate Center, Leibniz Institute for Primate Research, Göttingen, Germany

*For correspondence:
andreas.berghaenel@vetmeduni.ac.at

[†]These authors contributed equally to this work

Competing interest: The authors declare that no competing interests exist.

**Abstract** Adolescent growth spurts (GSs) in body length seem to be absent in non-human primates and are considered a distinct human trait. However, this distinction between present and absent length-GSs may reflect a mathematical artefact that makes it arbitrary. We first outline how scaling issues and inappropriate comparisons between length (linear) and weight (volume) growth rates result in misleading interpretations like the absence of length-GSs in non-human primates despite pronounced weight-GSs, or temporal delays between length- and weight-GSs. We then apply a scale-corrected approach to a comprehensive dataset on 258 zoo-housed bonobos that includes weight and length growth as well as several physiological markers related to growth and adolescence. We found pronounced GSs in body weight and length in both sexes. Weight and length growth trajectories corresponded with each other and with patterns of testosterone and insulin-like growth factor-binding protein 3 levels, resembling adolescent GSs in humans. We further re-interpreted published data of non-human primates, which showed that aligned GSs in weight and length exist not only in bonobos. Altogether, our results emphasize the importance of considering scaling laws when interpreting growth curves in general, and further show that pronounced, human-like adolescent length-GSs exist in bonobos and probably also many other non-human primates.

## eLife assessment

This **valuable** paper sheds new light on the growth trajectory of bonobos (*Pan paniscus*), with explicit contributions to discussions of the exclusivity of certain aspects of growth in modern humans, most specifically with respect to components of the adolescent growth spurt, which may be less human-specific among primates than presumed to this point. The results are **solid**, based on the largest sample ever considered in the study of bonobo growth and include both morphometric and endocrinological data. This work will be of interest to human evolutionary biologists, primatologists, and researchers studying the ontogeny and evolution of growth and development in general.

## Introduction

There is wide consensus that the human adolescent growth spurt (GS) in body length is evolutionary unique and absent in other primates (*Bogin, 2020*; *Gluckman et al., 2013*; *Hochberg, 2011*; *Holmgren, 2022*; *Stevens et al., 2013*; *Stulp and Barrett, 2016*, but see *Ellison et al., 2012*; *Sandel et al., 2023*; *Watts and Gavan, 1982*; *Weisfeld, 2006*). This is puzzling, because adolescent GSs in body weight occur in many primate species, including humans (*Leigh, 2001*). However, the apparent lack of an adolescent length-GS in non-human primates despite often pronounced weight-GSs may largely reflect a methodological problem that results from scaling issues when comparing length- and weight growth rates (*Cullen et al., 2021*; *Schmidt-Nielsen, 1984*). Here, we address this issue in three ways. First, we outline the pitfalls that result from such scaling issues, leading to e.g., failure to detect length-GSs even when these would perfectly and isometrically align with a massive weight-GS. Second, we use data from zoo-housed bonobos (*Pan paniscus*) to provide empirical proof of principle, showing how consideration of geometric length–weight scaling laws allows for more accurate detection of length-GSs. Moreover, our dataset includes longitudinal measures of body weight, forearm length, and muscle growth (proxied by creatinine levels) as well as measures of physiological markers of adrenarche (dehydroepiandrosterone, DHEA) and onset of sexual maturation (testosterone), and insulin-like growth factor-binding protein 3 (IGFBP-3) involved in the endocrinological regulation of the adolescent GS in humans (*Juul et al., 1995*; *Miller, 2022*). We investigated male and female growth patterns (weight, length, and muscle mass growth) and assigned them to developmental periods (changes in urinary DHEA, testosterone, and IGFBP-3 levels), with a particular focus on potential adolescent length-GSs. Third, we review previous literature on linear length growth in other non-human primates, and outline how the consideration of scaling laws may change their interpretation.

Growth trajectories can be highly variable in timing and amplitudes, leading to allometric and heterochronic differences between species, sexes, single tissues and body parts. Life history theory predicts that variance in somatic growth trajectories arises from allocation trade-offs between growth, reproduction, and self-maintenance, if investment in one trait comes at the expense of another (*Hau, 2007*; *Stearns, 1992*). Growing into large body size can increase survival, competitive power and somatic reproductive potential (*Flatt and Heyland, 2011*; *Stearns, 1992*). However, resource allocation to growth is limited by resource availability and traded-off against other developmental processes, and it can be beneficial to generally slow-down growth during early development and postpone it to right before reproductive maturation, resulting in adolescent GSs (*Leigh, 2001*; *Leigh, 1996*). Additionally, growth takes time and thus comes at the expense of a delayed onset of reproduction (*Flatt and Heyland, 2011*; *Stearns, 1992*). Therefore, optimal allocation strategies are expected to vary between sexes (*Berghänel et al., 2015*; *Hämäläinen et al., 2018*): For females, a long reproductive lifespan and thus an early onset of reproduction is crucial, whereas for polygynous males competitive power and a large adult body size during a relatively short period of prime status is more important (*Hämäläinen et al., 2018*; *Tarka et al., 2018*). Consequently, in polygynous species, males tend to be larger than females. Such sexual size dimorphism can be achieved by males either growing for a prolonged period and/or by growing at higher rates, which often leads to sex differences in the occurrence and pattern of adolescent GSs (*Leigh, 1996*).

The evolutionary origin and supposed uniqueness of the human adolescent GS remain disputed (*Bogin, 2020*; *Ellison et al., 2012*; *Gluckman et al., 2013*; *Holmgren, 2022*; *Sandel et al., 2023*; *Stevens et al., 2013*; *Stulp and Barrett, 2016*; *Watts and Gavan, 1982*; *Weisfeld, 2006*). Many primates, including humans and bonobos, show a GS in body weight in one or both sexes towards the end of their growth period, with differences in the occurrence, amplitude, timing, and/or duration of this weight-GS between species and between sexes (for detailed discussion of the evolution of such heterochrony and other variability [*Bogin, 2020*; *Leigh, 2001*; *Leigh and Shea, 1995*]). Such weight-GSs in non-human primates seem homologous to the adolescent weight-GS in humans, with the difference that in humans, chimpanzees (*Pan troglodytes*) and to some degree also bonobos, both adolescence and the associated weight-GS occur delayed compared to other non-human primates, and in humans seem also more squeezed (*Leigh, 2001*; *Leigh, 1996*). However, it has been argued that the human adolescent GS is nonetheless unique because beside the weight-GS, it also encompasses a strong GS in linear skeletal length that is evident in both sexes and often occurs even under unfavourable conditions (*Bogin, 2020*; *Hochberg, 2011*). In non-human primates, data on linear skeletal growth are scarce, but seem to indicate that accelerations in linear length growth are indeed

undetectable or marginal, even in species that show rather extreme weight-GSs such as mandrills (*Mandrillus sphinx*), chimpanzees (*P. troglodytes*), or gorillas (*Gorilla beringei beringei*) (*Galbany et al., 2017*; *Hamada and Udono, 2002*; *Setchell et al., 2001*; *Watts and Gavan, 1982*; but see *Berghänel et al., 2015*; *Lu et al., 2016*). This 'sharp contrast' (*Bogin, 2020*, p. 180) between weight- and length growth rates was recognized by both proponents and opponents of a unique human length-GS (*Bogin, 2020*; *Weisfeld, 2006*) and raised explanation attempts (*Gluckman et al., 2013*).

## Mind the scale: pitfalls in comparing weight- and length growth rates

One problem in interpreting somatic growth patterns in non-human primates and most other species may be that different measures for growth rates are compared with one another, including weight- and length growth rates (*Figure 1*). The problem arises from a scaling issue and an inappropriate comparison between linear (length) and cubic (volume = weight) growth (*Cullen et al., 2021*; *Schmidt-Nielsen, 1984*). In a simplified case of isometric weight growth, length growth would follow a cubic-root function of weight growth to align, an aspect that is generally acknowledged and underlies discussions about weight–height ratios like the body mass index (*Cullen et al., 2021*; *Schmidt-Nielsen, 1984*). Comparisons and interpretations of weight- and length growth rates then typically build on two assumptions. First, it is correctly assumed that notwithstanding their non-linear relationship, weight- and length values still rise and fall together. Second, it is incorrectly assumed that the same logic also applies to the respective growth rates (*Figure 1*; *Cullen et al., 2021*). The cubic relationship between weight and length growth inevitably results in a decreasing or, at best, constant length growth rate if the acceleration in weight growth rate does not exceed a quadratic function of age (*Figure 1*; *Cullen et al., 2021*). Hence, in many cases, an acceleration in weight growth rate (=spurt) would be accompanied by a decreasing length growth rate, even if both are perfectly aligned with each other in isometric growth. As previously noticed (e.g., *Hamada and Udono, 2002*), a length-GS is only detectable if the actual acceleration resulting from the weight-GS exceeds the parallel deceleration in length growth rate that results from the cubic relationship. In fact, whether a linear length-GS is detectable is largely independent of the amplitude of the associated weight-GS, and rather depends on various other aspects, such as how fast weight growth rate accelerates, how the level of acceleration changes (e.g., linear or quadratic acceleration), or at which body size the acceleration takes off (for details see *Figure 1*). Therefore, age trajectories of linear length growth rate are difficult to interpret and require consideration of many aspects. Consequently, a dichotomy between present or absent acceleration in linear length growth seems somewhat arbitrary and leads to misinterpretations (*Cullen et al., 2021*).

Ignoring the scaling issue can cause several methodological artefacts, including (1) undetectable length-GSs despite pronounced weight-GSs, (2) evidence for length-GSs in larger but not small species despite similar weight-GSs, and (3) a pronounced time lag between length- and weight-GS, even if both are perfectly synchronous (for detailed explanation see *Figure 1*). Additionally, detectability changes with increasing weight. On the one hand, (4) the same acceleration in weight becomes increasingly detectable in linear length (*Figure 1F*), but on the other hand, (5) the same difference in absolute body weight is accompanied by smaller and smaller differences in linear length (*Figure 1A*), which makes absolute differences in length more difficult to detect. Importantly, the resource-limited aspect of growth and thus, the dimension of interest for life history trade-offs is biomass production, and therefore, weight growth (*Leigh, 1996*; *Schmidt-Nielsen, 1984*), which then may also mediate limits on length growth. Hence, even though body length may itself be under selection, its investigation within a life history framework is only reasonable if its scaling and alignment with body weight is considered.

## Mind the life stage: is the growth spurt an adolescent growth spurt

Another problem with detecting adolescent GSs is how growth patterns are assigned to developmental stages like adolescence. In non-human primates, developmental stages or events are sometimes vaguely defined or based on proxy measures such as dental eruption or epiphyseal closing, which is usually impossible to assess in vivo (*Bolter and Zihlman, 2012*; *Gavan, 1953*). Physiological markers offer an alternative method to differentiate developmental stages in both sexes, and may be more sensitive in detecting age-related changes (*Miller, 2022*). Physiological systems have been proposed as the major regulatory mechanisms adjusting energy trade-offs and allocation, and thus,

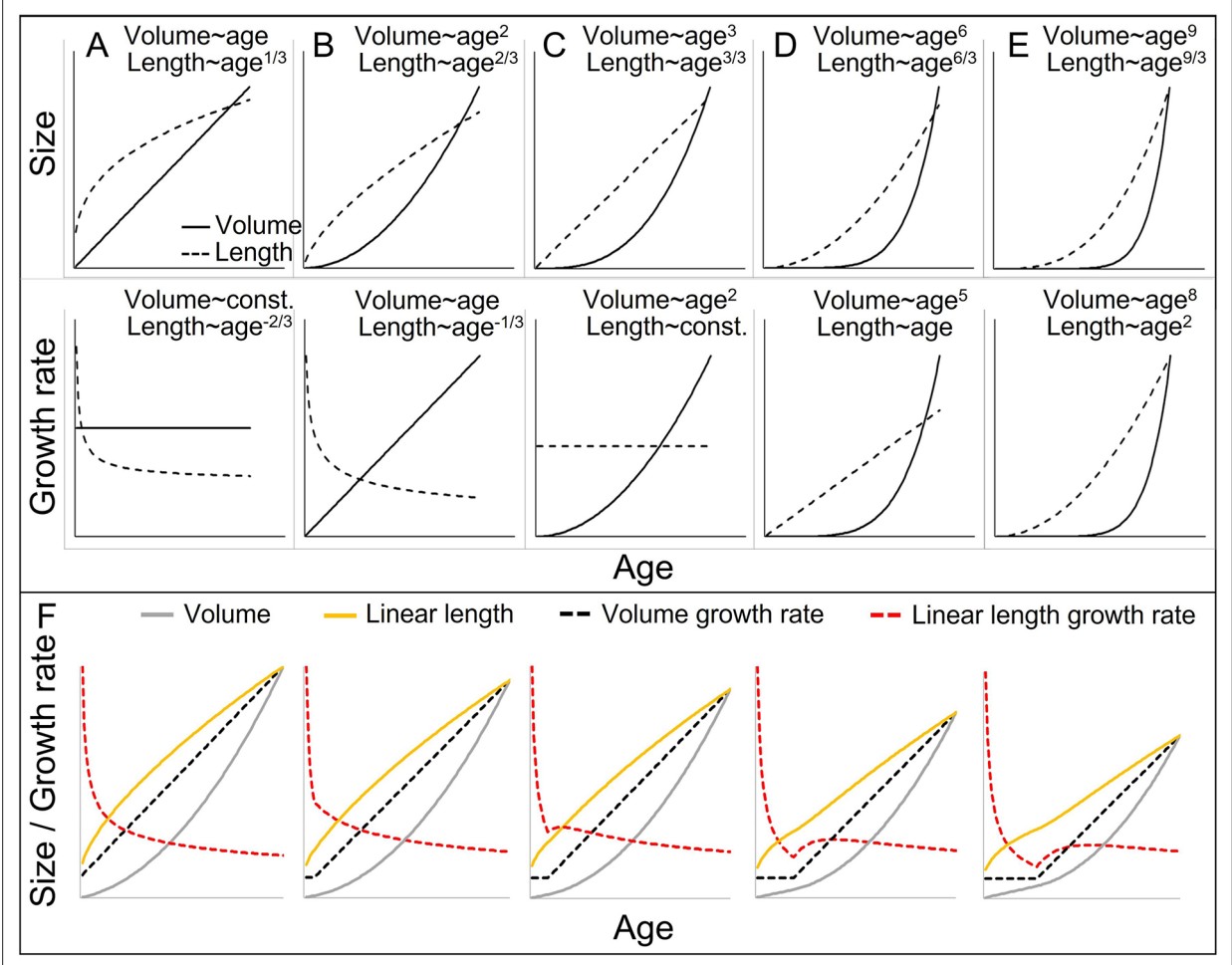

**Figure 1.** Cubic isometric relationship between volume (~weight) and length growth, and likelihood to detect an existing growth spurt (GS) in linear length (schematic). (**A–E**) Top/bottom: Absolute size and growth rate (=1st derivation of size). From left to right: Increasingly fast acceleration of volume (~weight) growth and the aligned length growth, from (**A**) no acceleration (constant volume growth rate and linear increase in volume) through (**B**) constant acceleration (linear increase in volume growth rate and quadratic increase in volume) to (**C**) quadratic acceleration of volume growth rate (cubic increase in volume), and (**D, E**) even faster volume growth acceleration. Due to the cubic relationship, these volume growth rates would align with decreasing (**A, B**) or constant length growth rates (**C**), whereas a detectable acceleration in length growth rate may only be found in cases of very fast acceleration in volume growth rate (**D, E**). Therefore, the current dichotomy between absent and detectable length GSs would only differentiate between (**A–C**) and (**D, E**). Another consequence is that, in non-aquatic animals, a cubic relationship is more likely in smaller animals, whereas in larger animals like humans or bonobos, the relationship tends to follow a lower power of 2.5 or even 2 only, as a result from limitation on the bearable weight of a skeletal construction which relates to the sectional area of bones (for more details see e.g., *Juul et al., 1995*). This means that in case of an equal volume growth acceleration, an aligned acceleration in linear length growth may become more likely detectable in larger animals simply because of the different underlying scaling laws. (**F**) The above scaling rules lead to further dynamics depending on the temporal overlap of the curves, making length GSs more pronounced and detectable with increasing size (from left to right). A GS in linear length is detectable if the acceleration resulting from the volume-GS exceeds the deceleration in length growth rate that results from the cubic relationship, with the last one becoming weaker with increasing size, respective age. The figure shows how a change from a constant to a linearly accelerating volume growth rate (like in **A and B**; equal levels of acceleration) results in different levels of acceleration in linear length growth rate depending on the age/size at which this change occurs, from left (change right after birth, only deceleration in length growth rate) (equal to **B**) to right (change at late age, strong acceleration in length growth rate). Additionally, this figure highlights that even if both volume and linear length show a GS and are perfectly aligned, the linear length growth rate reaches its peak and starts declining again at a time when volume growth rate still increases. See also *Figure 1—figure supplement 1* for non-linear acceleration in volume growth rate.

The online version of this article includes the following figure supplement(s) for figure 1:

**Figure supplement 1.** The figure shows how a change from a constant to a quadratically accelerating volume growth rate at a certain body size (identical levels of acceleration) results in different levels of acceleration in linear length growth rate depending on the age (in fact, body size) at which this change occurs, from left (change right after birth at small body size, only slight acceleration in linear length growth rate) to right (change at late age and larger body size, strong acceleration in length growth rate).

adaptively coordinating the expression of life history traits (*Del Giudice, 2020*). Thus, physiological makers allow to monitor specific life history transitions at a more detailed level, and facilitate inter-specific comparison.

In the empirical part of our study, we used three physiological markers that are indicative of life history stages, but are also directly related to growth. First, DHEA and DHEA-sulfate (DHEA-S) levels allow to determine the onset of adrenarche, the maturation of the adrenal cortex and the onset of adrenal androgen secretion. This is one of the first life history events during postnatal development characterizing the onset of the juvenile period. Increasing levels of DHEA(-S) might be involved in brain development in apes and humans, and inhibit long-bone growth in humans, paralleling the low growth rates during the juvenile period (*Hochberg, 2011*). Second, changes in testosterone levels indicate testicular maturation in boys and ovarian maturation in girls, and are therefore a physiological marker for the onset of sexual maturation and associated with the adolescence stage in both sexes (*Bribiescas, 2010*; *Muller, 2017*). Furthermore, testosterone promotes muscle growth (*Bribiescas, 2010*). In boys, the adolescent GS in muscle mass lasts longer than the skeletal GS, with muscle growth proceeding into early adulthood (*Bogin, 2020*). Third, insulin-like growth factor 1 (IGF-1) binds to IGFBP-3 for transportation, and levels of both increase with increasing levels of testosterone. Hence in primates, IGFBP-3 levels increase strongly at the beginning of puberty, and are directly linked to rates of length growth and, in combination with testosterone, to muscle growth during the adolescent GS (*Bernstein et al., 2008*; *Juul et al., 1995*; *Miller, 2022*).

## Growth in bonobos: a comprehensive test case

Our empirical data combined measures of somatic growth with measures of physiological markers to investigate adolescent GSs in bonobos. Information about the life history and development of bonobos is still sparse (*Behringer et al., 2016a*; *Jungers and Susman, 1984*; *Leigh, 1996*). In wild-living females, genital swellings start to increase in size around 5–6 years of age (*Kano, 1992*), which is also when female testosterone levels rise in captivity (*Behringer et al., 2014*). Females disperse from their natal group between 5 and 9 years (*Toda et al., 2022*). In zoo-housed bonobos, menarche occurs between the age of 6 and 11.3 years (*Thompson-Handler, 1990*; *Vervaecke et al., 1999*), and they give birth for the first time around 10.7 ± 3.3 years (range: 8–15 years), which is significantly earlier compared to wild populations, where females give first birth around 14.2 years of age (*De Lathouwers and Van Elsacker, 2006*). In males, testicular descent occurs at 9 years of age in wild bonobos (*Kuroda, 1989*) and between the sixth and tenth year of age in captivity (*Dahl and Gould, 1997*), which corresponds to a fast increase in urinary testosterone levels around the age of 8 years (*Behringer et al., 2014*). Zoo-born males are on average 12.3 years old at their first reproduction (range 7–17.2 years) (*Reinartz et al., 2002*). Zoo-housed females outlive males by several decades (*Stevens, 2020*). Compared to other hominoid primates, sexual dimorphism in adult body weight is exceptionally small in bonobos (*Leigh and Shea, 1995*), and preliminary data on linear dimensions such as postcranial skeleton, body segments, and forearm length suggest a complete absence of sexual size dimorphism in body length (*Behringer et al., 2016a*; *Druelle et al., 2018*). In some monomorphic species, males and females do grow at the same rate for a similar duration, whereas others acquire adult body size by bimaturation (*O'Mara et al., 2012*). Thus, the low extent of adult sex dimorphism alone is a poor predictor for somatic growth trajectories. In bonobos, growth patterns have so far been studied on limited data. For example, based on 13 males and 23 females, it was suggested that both male and female bonobos experience a subadult GS in body weight (*Leigh, 1996*). So far, no data on linear growth trajectories with corresponding physiological changes have been published.

Here, we provide a large and comprehensive dataset on patterns of growth and physiological ontogeny in zoo-housed bonobos. This dataset allows to critically test the hypothesis that human-like adolescent length-GSs are absent in other primates. We investigated sex-specific GSs in body weight, forearm length, and muscle mass measured as urinary creatinine corrected for specific gravity (*Emery Thompson et al., 2012*). We combined these three measures of growth with three physiological markers in urine for the timing of developmental stages: DHEA as a marker for adrenarche, testosterone as a marker for onset of sexual maturation, and IGFBP-3 as a marker for the adolescent GS. We first investigated whether GSs in body weight, forearm length, or muscle mass are evident in either sex, and how they are aligned with each other if compared in the same dimension (i.e., after correction for scaling, *Figure 1*). Then we tested whether these potential GSs relate to physiological markers for

the juvenile or the adolescent period (increase in urinary DHEA or testosterone, respectively), and whether they align with increasing levels of IGFBP-3 which was shown to directly relate to adolescent GSs in humans (*Juul et al., 1995*; *Miller, 2022*).

## Other primates: is there evidence for pronounced GSs in length?

In a last step, we did a non-systematic literature search on studies investigating length growth rates in other non-human primates. In addition to searching for linear length-GS, we also searched for other patterns that would be likely to show a length-GS if investigated at the scale-corrected dimension. Primarily, periods of constant, plateauing growth rates in linear length do very likely represent length-GSs as can be seen in *Figure 1C*. Furthermore, studies showing a slowdown in the deceleration in linear length growth might be promising candidates. In addition, we searched for known male and female markers of adolescence to estimate whether GSs would be aligned with such markers and could be assigned to adolescence.

## Results

### Growth trajectories in bonobos

We applied Generalized Additive Mixed Modelling (GAMM) to our dataset including a comprehensive non-linear random effects structure that implements variation in age trajectories between individuals, between rearing conditions, and between zoos, plus variation resulting from zoo-specific changes in conditions over the years (for details see methods section and *Table 1*). This approach provided realistic confidence intervals of fitted values and curves, and thus, of variability and uncertainty in the occurrence, timing, and magnitude of patterns like potential GSs (*Pedersen et al., 2019*; *Wieling, 2018*; *Wood, 2017*).

We further re-ran our analyses on two reduced – and more conservative – datasets (see methods section and supplemental material). Our dataset included several wild-born individuals. For those individuals, some information on e.g., exact birthdate, parents or early life conditions that may have influenced their developmental trajectories were not available. Therefore, we re-run all our analyses excluding wild-born individuals, which allowed for controlling for parental ID (sire and dam) and maternal age. Moreover, due to the long-term sampling effort, corresponding data on body weight, forearm length as well as DHEA and testosterone levels were not always available for each individual, and IGFBP-3 was analysed only in some urine samples because of a small urine volume. Therefore, we re-run our analyses on body weight, forearm length, DHEA, and testosterone levels on the subset of individuals for which data on all four measures were available. These analyses yielded identical patterns as the full model (*Figure 2—figure supplement 1*, *Figure 3—figure supplements 1 and 2*).

On average, adult male bonobos were significantly larger, heavier and had higher lean body mass than females, but there was also a wide overlap between sexes (*Figure 2*). Both males and females showed a pronounced GS in body weight (*Figure 2A*) that was accompanied by a similarly pronounced GS in forearm length which almost perfectly aligned in timing and amplitude (*Figure 2B* and 4A). Compared to females, male bonobos reached peak growth velocity in both body weight and forearm length 2 years later (6 vs. 8 years of age), and had a longer and in case of weight growth also more pronounced GS, which resulted in a larger adult body size in males (*Figure 2A, B*). There was no period of decelerated growth prior to this GS in either sex.

The pronounced GS in male forearm length was only evident if analysed at the weight dimension (i.e., raised to the power of $cm^{2.5}$, *Figure 2B*). Remarkably though, the GS in female forearm length was also evident if investigated at linear length (in cm/year, *Figure 2D*). However, female length growth reached its peak velocity about 1 year earlier when reported in cm/year compared to $cm^{2.5}$/year (Figure 4A, red and blue dotted line), as predicted based on the mathematical scaling relationship (*Figure 1*). Consequently, only the corrected GS (in $cm^{2.5}$/year) matched the GS in body weight (Figure 4A). Reversely, if down-scaling body weight to the dimension of linear length growth (i.e., extracting the 2.5th root, *Figure 2C*), the pronounced weight-GSs were no longer detectable, and the resulting pattern matched the trajectories in linear length growth (in cm, *Figure 2D*).

We also found a GS in lean body respective muscle mass in males (*Figure 3A*), which was 1.5 years after the length- and the weight-GS (Figure 4A), though the difference in timing was within the 95%

**Table 1.** Statistical results of Generalized Additive Mixed Models (GAMMs) on growth and physiology.

Blue: Interaction term results from a separate model (see methods section). Red: Special model structure for insulin-like growth factor-binding protein 3 (IGFBP-3) models (random intercept per individual, random smooth per zoo not sex specific; for details see methods). §: including maternal primiparity, rearing conditions (hand- vs. mother-reared) and zoo- vs. wild-born (see methods section). *: $\sqrt[2.5]{kg}$. Model p-values result from null model comparison. Est. = Estimate.

| Factor variables | Reference Category | Body weight (kg) | | | | Body weight ($\sqrt{kg}$*) | | | | Lower arm length (cm) | | | | Lower arm length (cm$^{2.5}$) | | | |
|---|---|---|---|---|---|---|---|---|---|---|---|---|---|---|---|---|---|
| | | Est. | SE | t | p | Est. | SE | t | p | Est. | SE | t | p | Est. | SE | t | p |
| (Intercept) | | 22.4 | 0.10 | 222 | <0.001 | 3.17 | 0.01 | 581 | <0.001 | 25.2 | 0.06 | 443 | <0.001 | 3504 | 19.1 | 183 | <0.001 |
| Males | Females | 4.10 | 0.13 | 30.8 | <0.001 | 0.20 | 0.01 | 26.9 | <0.001 | 1.26 | 0.19 | 6.77 | <0.001 | 567 | 63.0 | 9.00 | <0.001 |
| **Smooth term variables** | | edf | Ref.df | F | p | edf | Ref.df | F | p | edf | Ref.df | F | p | edf | Ref.df | F | p |
| *Age trajectories* | | | | | | | | | | | | | | | | | |
| Females | | 9.12 | 9.39 | 82.3 | <0.001 | 9.00 | 9.19 | 52.1 | <0.001 | 8.32 | 8.71 | 120 | <0.001 | 8.44 | 8.78 | 54.7 | <0.001 |
| Males | | 9.53 | 9.67 | 116 | <0.001 | 9.42 | 9.52 | 52.4 | <0.001 | 7.75 | 8.37 | 115 | <0.001 | 8.06 | 8.56 | 43.2 | <0.001 |
| Males | Females | 8.53 | 8.84 | 6.36 | <0.001 | 9.04 | 9.26 | 7.42 | <0.001 | 7.05 | 7.77 | 5.34 | <0.001 | 6.71 | 7.48 | 4.49 | <0.001 |
| *Random smooths* | | | | | | | | | | | | | | | | | |
| Sampling date per zoo | | 103 | 175 | 2.85 | <0.001 | 97.6 | 175 | 2.53 | <0.001 | 10.7 | 44.0 | 0.49 | <0.001 | 7.05 | 37.0 | 0.47 | <0.001 |
| Age trajectory per individual | | 600 | 1544 | 14.4 | <0.001 | 577 | 1545 | 10.5 | <0.001 | 148 | 336 | 14.6 | <0.001 | 151 | 336 | 20.5 | <0.001 |
| Age trajectory per rearing§ | | 2.12 | 53.0 | 0.12 | <0.001 | 27.6 | 53.0 | 2.29 | <0.001 | 0.00 | 48.0 | 0.00 | 0.116 | 0.00 | 48.0 | 0.00 | 0.022 |
| Female age trajectory per zoo | | 32.1 | 175 | 0.29 | <0.001 | 21.9 | 175 | 0.18 | <0.001 | 0.01 | 23.0 | 0.00 | 0.070 | 0.02 | 23.0 | 0.00 | 0.055 |
| Male age trajectory per zoo | | 35.1 | 168 | 0.35 | <0.001 | 61.0 | 168 | 0.91 | <0.001 | 6.69 | 26.0 | 0.88 | <0.001 | 10.9 | 26.0 | 1.45 | <0.001 |
| $R^2_{adj}$ (deviance explained) | | 0.995 (99.5%) | | | | 0.997 (99.7%) | | | | 0.988 (99.1%) | | | | 0.986 (99.0%) | | | |
| N (p-value) | | 8355 (<0.001) | | | | 8355 (<0.001) | | | | 641 (<0.001) | | | | 641 (<0.001) | | | |

| Factor variables | Reference Category | Ln(Creatinine) | | | | Ln(DHEA) | | | | Ln(Testosterone) | | | | Ln(IGFBP-3) | | | |
|---|---|---|---|---|---|---|---|---|---|---|---|---|---|---|---|---|---|
| | | Est. | SE | t | p | Est. | SE | t | p | Est. | SE | t | p | Est. | SE | t | p |
| (Intercept) | | –0.02 | 0.07 | –0.96 | 0.337 | 2.99 | 0.04 | 84.2 | <0.001 | 0.43 | 0.05 | 9.15 | <0.001 | 2.06 | 0.08 | 25.6 | <0.001 |
| Males | Females | 0.14 | 0.04 | 3.97 | <0.001 | –0.05 | 0.04 | –1.09 | 0.276 | 0.97 | 0.10 | 9.34 | <0.001 | 0.04 | 0.12 | 0.34 | 0.732 |
| **Smooth term variables** | | edf | Ref.df | F | p | edf | Ref.df | F | p | edf | Ref.df | F | p | edf | Ref.df | F | p |
| Daytime | | 1.00 | 1.00 | 8.98 | 0.003 | 1.77 | 1.94 | 2.04 | 0.116 | 1.00 | 1.00 | 2.46 | 0.118 | 1.00 | 1.00 | 2.94 | 0.089 |
| *Age trajectories* | | | | | | | | | | | | | | | | | |
| Females | | 2.64 | 2.99 | 6.17 | <0.001 | 4.69 | 5.44 | 6.16 | <0.001 | 8.48 | 8.85 | 19.2 | <0.001 | 5.04 | 5.65 | 5.39 | <0.001 |
| Males | | 4.45 | 4.77 | 20.4 | <0.001 | 2.97 | 3.38 | 9.77 | <0.001 | 8.33 | 8.81 | 30.6 | <0.001 | 3.98 | 4.66 | 5.53 | <0.001 |
| Males | Females | 4.41 | 4.73 | 3.73 | 0.016 | 1.00 | 1.00 | 0.24 | 0.622 | 8.05 | 8.63 | 8.24 | <0.001 | 3.24 | 3.62 | 3.27 | 0.010 |
| *Random smooths* | | | | | | | | | | | | | | | | | |
| Sampling date per zoo | | 24.7 | 56.0 | 2.61 | <0.001 | 33.0 | 55.0 | 10.5 | <0.001 | 25.4 | 58.0 | 3.46 | <0.001 | - | - | - | - |
| Age trajectory per individual | | 42.0 | 385 | 0.18 | <0.001 | 14.88 | 392 | 0.05 | 0.031 | 66.4 | 397 | 0.38 | <0.001 | 0.00 | 103 | 0.00 | 0.819 |
| Age trajectory per rearing§ | | 0.00 | 28.0 | 0.00 | 0.191 | 0.00 | 28.0 | 0.00 | 0.651 | 1.79 | 28.0 | 0.24 | <0.001 | - | - | - | - |
| Female age trajectory per zoo | | 7.55 | 34.0 | 0.57 | <0.001 | 2.77 | 60.0 | 0.06 | 0.042 | 6.66 | 66.0 | 0.26 | <0.001 | 0.00 | 35.0 | 0.00 | 0.554 |
| Male age trajectory per zoo | | 0.00 | 33.0 | 0.00 | 0.087 | 0.58 | 60.0 | 0.01 | 0.13 | 2.46 | 63.0 | 0.05 | 0.006 | - | - | - | - |
| $R^2_{adj}$ (deviance explained) | | 0.424 (48.7%) | | | | 0.488 (52.7%) | | | | 0.712 (75.5%) | | | | 0.235 (28.7%) | | | |
| N (p-value) | | 766 (<0.001) | | | | 782 (0.001) | | | | 802 (<0.001) | | | | 163 (0.003) | | | |

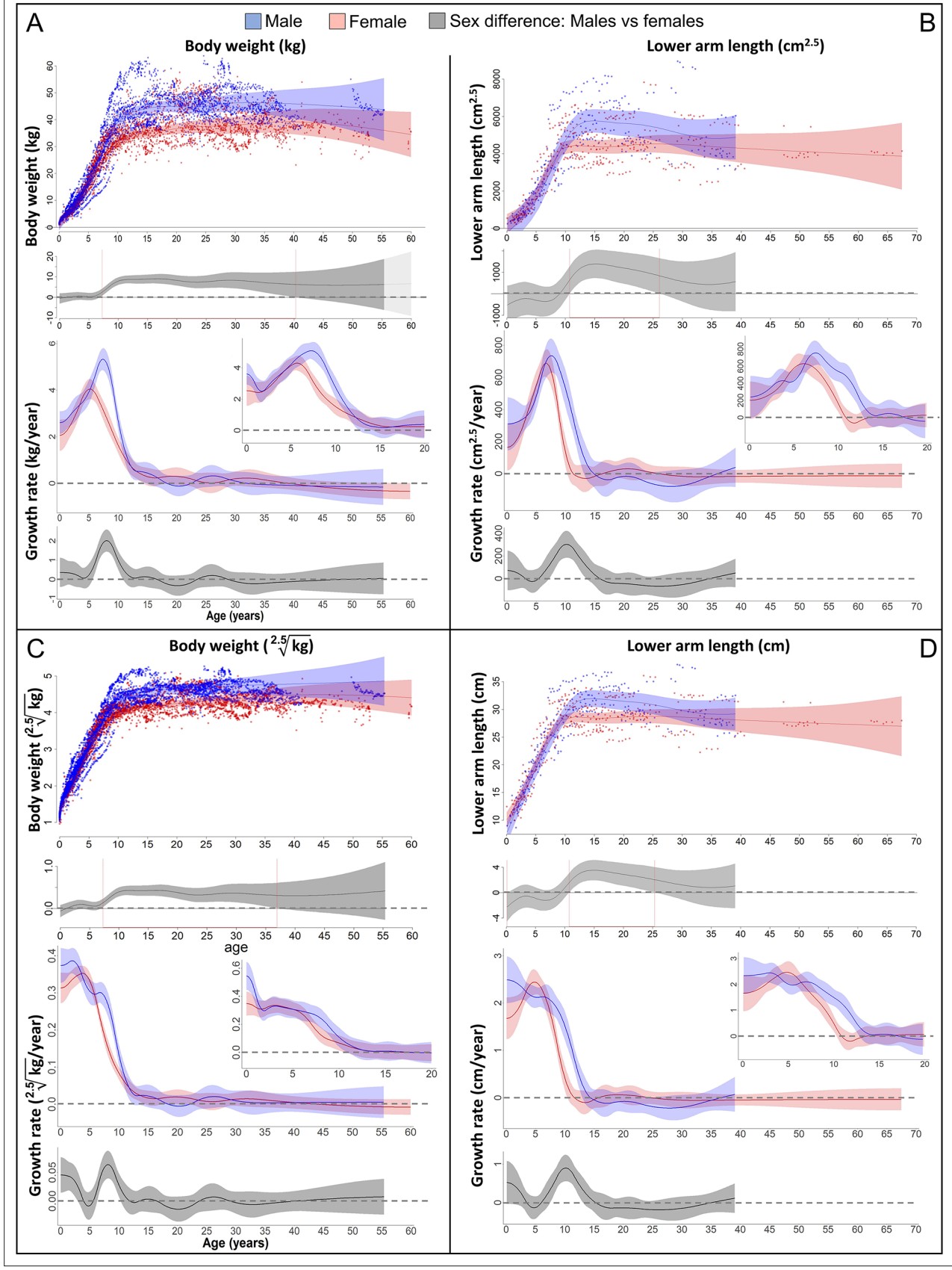

Figure 2 continued on next page

*Figure 2 continued*

**Figure 2.** Growth trajectories in body weight and forearm length, and the importance of comparing them at the relevant dimension. Fitted values and 95% CIs from Generalized Additive Mixed Models (GAMMs) are shown, implementing variability in trajectories across individuals and zoos. (**A, B**) Investigating both weight and length growth at the dimension of weight growth reveals pronounced growth spurts (GSs) in both which are strongly aligned with each other in timing and amplitude (see also *Figure 4A* for easier comparison). (**C, D**) If instead examined at the scale of one-dimensional length growth, weight, and length growth curves still align with each other but the GSs are not so easily detectable anymore. However, the GS is still evident in female growth, but appears at a younger age than if scale corrected (**A, B**). The potential fast decrease in growth rate after birth was not covered by linear length data (first half year of age: 2 arm and 854 [729 male] weight measures).

The online version of this article includes the following figure supplement(s) for figure 2:

**Figure supplement 1.** Our results on body weight (left) and forearm length (right) remained the same if (**A, B**) only zoo-born individuals were considered (which also allowed to additionally control for kinship [dam and sire] and maternal age at birth) or if (**C, D**) only those individuals were considered for which data on body weight, forearm length, dehydroepiandrosterone (DHEA), and testosterone were available.

**Figure supplement 2.** Validation of the scaling relationship between forearm length and body weight during growth in our study.

CIs and not significant. In females, we found no particular peak in the lean body mass growth rate, but growth rate was highest until 5 years of age and decreased thereafter (*Figure 3A*).

In both sexes, DHEA levels increased fastest during the first 5 years and reached maximum levels around 15 years of age, with similar levels and age trajectories (*Figure 3B*). As predicted for an adolescent GS, growth patterns and DHEA levels were not associated, and DHEA levels did not show any sex difference in levels or age trajectories that would align to the sex differences in growth (*Figure 4*).

In both males and females, testosterone and IGFBP-3 levels increased largely together (within the wide range of 95% CIs particularly for IGFBP-3) but in a sex-specific manner, rising at an earlier age in females than males (*Figures 3C, D and 4B*). IGFBP-3 levels reached similar maximal levels in both sexes and declined thereafter (*Figure 3D*), whereas testosterone levels increased for longer and reached higher levels in males than females, and remained at the maximum level for many years (*Figure 3C*).

Testosterone levels increased fast at the age of GS take-off in both sexes (*Figure 4*). Testosterone reached maximum levels already before the age of peak growth velocity in females, but slightly after the peak in length- and weight-growth and more closely to the peak in muscle-growth in males.

The weight- and length-GSs were largely aligned with IGFBP-3 levels (*Figure 4*). In females, IGFBP-3 levels directly aligned with the GS in length and weight and peaked at the same age, whereas in males they reached their maximum levels after the peak in weight and length growth and better matched changes in muscle growth rates, also after considering the wide range of 95% CIs for IGFBP-3. DHEA and testosterone levels declined after 30 years of age (*Figure 3B, C*).

## Other primate studies on weight and length growth

We found 13 primate species for which data on length growth rate over age were available (*Table 2*). For nine species, there was evidence for an acceleration in linear length growth in at least one sex, with all other species showing a period of constant length growth rate or a slowdown in the deceleration. For two species, data on scale-corrected (cubic-transformed) growth were available. In wild Assamese macaques (*Macaca assamensis*), linear length growth rate only showed a slowdown in deceleration whereas cubic-transformed values showed a significant acceleration in length growth, which occurred also at a later age then the slowdown in linear length growth. In Pigtailed macaques (*Macaca nemestrina*), cubic-root-transformed weight measures were perfectly linearly correlated with length values. For 10 species for which parallel weight growth data were available, the respective pattern in linear length coincided with a weight-GS, but as predicted, this linear pattern appeared at younger ages than the weight-GS in all species where available data allowed for such detailed analyses. Furthermore, for the 13 primate species, we searched for published proxies of the age of adolescence, such as increase in testes size and testosterone levels in males, and age at first swelling, cycling, ovulation, or menarche in females (*Appendix 1—table 1*). In most cases, these data were derived from different study populations, but the data on adolescence and growth were both from the same environment, wild, or captive. Overall, GSs occurred at similar ages as reported markers of adolescence within a species, but this comparison was often coarse and non-conclusive, as e.g., age at menarche or first cycling was highly variable for females of most ape species, and increase in male

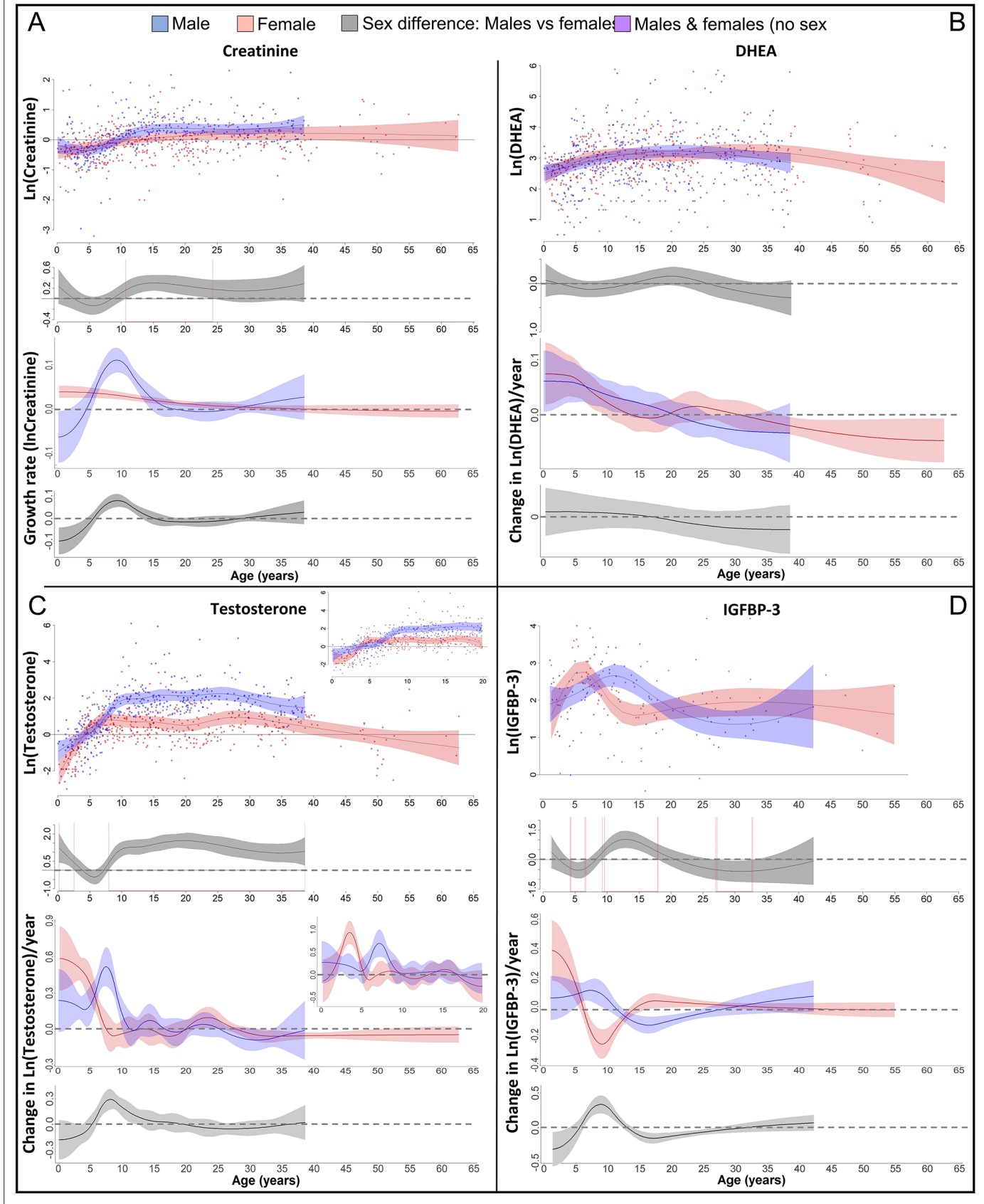

Figure 3 continued on next page

*Figure 3 continued*

**Figure 3.** Physiological changes during ontogeny: markers of muscle growth (creatinine), adrenarche (dehydroepiandrosterone, DHEA), and adolescence (testosterone and insulin-like growth factor-binding protein 3 [IGFBP-3]). Fitted values and 95% CIs from Generalized Additive Mixed Models (GAMMs) are shown, which implement variability in trajectories across individuals and zoos. (**A**) Males showed a pronounced growth spurt (GS) in lean body respective muscle mass, resulting in larger lean body mass in males compared to females where such a GS was not detectable. Be aware though that corrected creatinine values before the age of 3–4 years may be less reliable (*Emery Thompson et al., 2012*). (**B**) DHEA levels increased fastest during the first 5 years of life and reached maximal levels at ~15 years. (**C**) Testosterone levels increased during development in both sexes, but in males, testosterone levels increased longer and reached higher adult levels. They increased fastest at 3.5–4 years in females and 7 years in males. Testosterone levels decreased again after the age of ~30 years of age. (**D**) IGFBP-3 levels showed a peak of similar height in males and females, occurring at a younger age in females than males.

The online version of this article includes the following figure supplement(s) for figure 3:

**Figure supplement 1.** Our results on dehydroepiandrosterone (DHEA) (left) and testosterone (right) remained the same if (**A, B**) only zoo-born individuals were considered (which also allowed to additionally control for kinship [dam and sire] and maternal age at birth) or if (**C, D**) only those individuals were considered for which data on body weight, forearm length, DHEA, and testosterone were available.

**Figure supplement 2.** Our results on creatinine (**A**) and insulin-like growth factor-binding protein 3 (IGFBP-3) (**B**) remained the same if only zoo-born individuals were considered (which also allowed to additionally control for kinship [dam and sire] and maternal age at birth).

**Figure supplement 3.** In our raw data, females and males showed a fast increase in testosterone, as also previously shown for bonobos (*Behringer et al., 2014*), with females (red) reaching maximal levels around the age of 5, and males (blue) around the age of 9 years.

testes size and testosterone levels were often aligned with, but sometimes also occurred at later ages than the GSs (*Appendix 1—table 1*).

## Discussion
### Growth trajectories and adolescent GSs in zoo-housed bonobos

Comparing weight and length growth at comparable scales revealed a pronounced and concerted GS in weight and length in male and female bonobos, with the length-GS in females being also evident at one-dimensional linear length growth scale. Our results on weight growth and adult sex difference in body weight largely match previous results (*Leigh, 1996*), but peak velocity occurred 1 year later in both sexes in our study, indicating a stronger and more human-like delay of the adolescent GS than previously assumed (*Leigh, 2001*; *Leigh, 1996*). In contrast to previous results on much smaller sample sizes (*Behringer et al., 2016a*; *Druelle et al., 2018*) we also found an adult sex difference in forearm length, indicating a sexual size dimorphism in length in bonobos.

The aligned weight- and length-GS was further accompanied by a slightly delayed GS in lean body mass in males, but not females, a pattern similar to the muscle GS in humans (*Bogin, 2020*). However, a muscle GS in females may have been masked because creatinine measurement can show high uncertainty before the age of 3 years (*Emery Thompson et al., 2012*), hence our result on sex-specific muscle growth in bonobos should be interpreted with caution and needs further validation.

One advantage of our bonobo study was the ability to determine adolescence through physiological measures, allowing to differentiate whether the observed GSs could be assigned to adolescence. Both the male and the female GS in weight and length corresponded to sex-specific increase in testosterone levels, with matching ages at rise take-off. In females, the peak of the GS in length and weight coincided with maximal testosterone and IGFBP-3 levels, whereas in males, maximal testosterone and IGFBP-3 levels were more related to the delayed GS in and the maintenance of muscle mass. Our results on testosterone trajectories confirm the previous finding that testosterone levels reach maximal values at a much younger age in female bonobos compared to male bonobos (*Behringer et al., 2014*). In females, testosterone levels showed a fast rise at about 4 years of age and reached maximal levels at 5 years of age, matching previous results on testosterone level changes (*Behringer et al., 2014*), menarche at 6–11 years of age in zoo-housed females (*Thompson-Handler, 1990*; *Vervaecke et al., 1999*), and increasing external genitalia with 5–6 years of age in wild female bonobos (*Kano, 1992*). In males, testosterone levels showed a fast increase at about 7 years of age and reached maximal levels at about 9 years of age, which is in line with previous results on testosterone level changes (*Behringer et al., 2014*), testicular descent at 9 years of age in wild (*Kuroda, 1989*), and between 6 and 10 years of age in zoo-housed male bonobos (*Dahl and Gould, 1997*). It will be important to research the more complex hormonal underpinnings in the future, which would

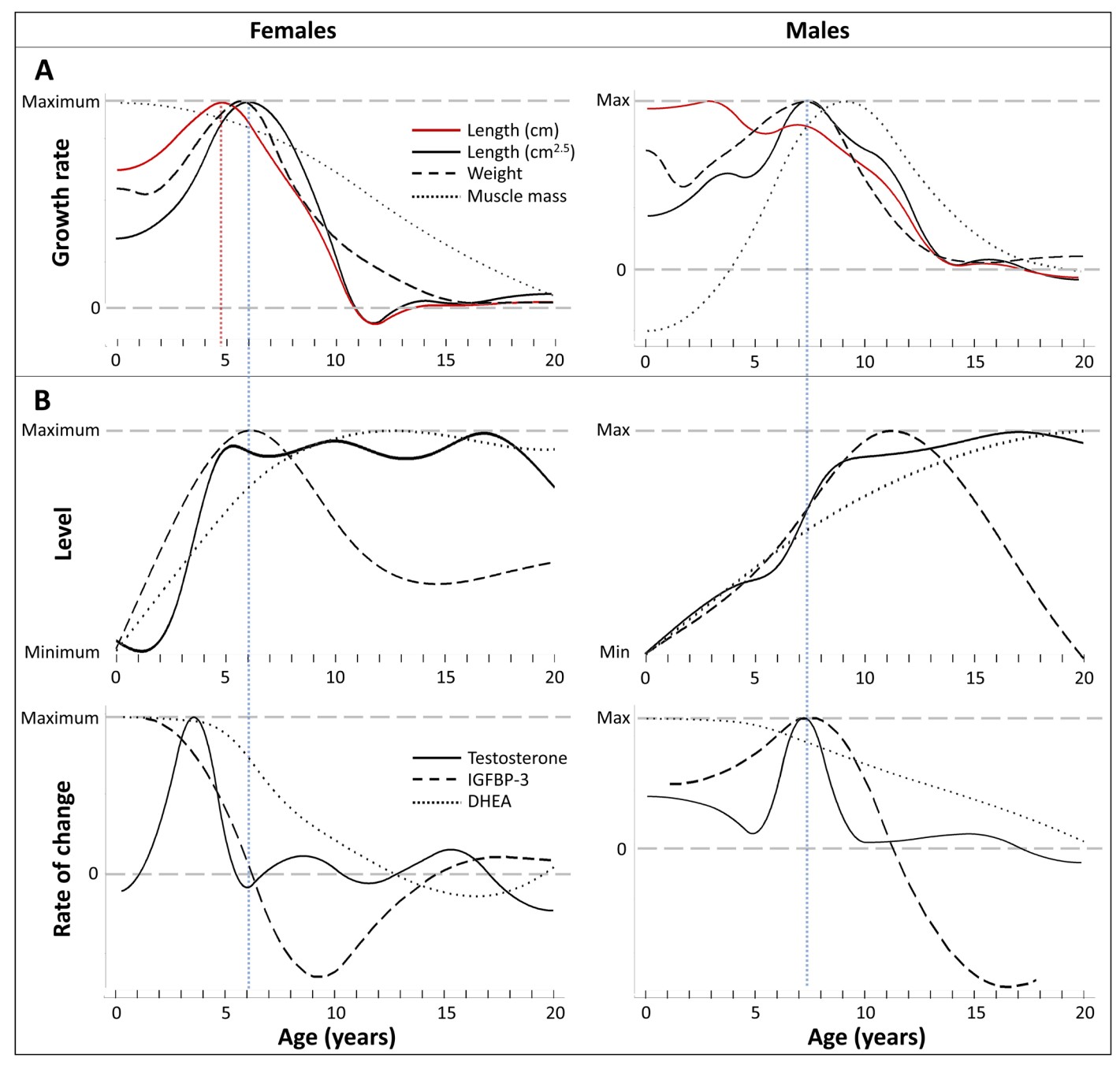

**Figure 4.** Direct comparison of age trajectories in growth patterns and physiological parameters until the age of 20 years. All curves are the same as in *Figures 2 and 3*, and for the respective variability and uncertainty in the trajectories including the occurrence, level, and timing of peaks see the 95% CIs in *Figures 2 and 3*. Blue and red dotted line: Age at peak growth velocity in $cm^{2.5}$/year (blue) and in cm/year (red; females only). (**A**) Change of growth rate over age in weight, forearm length, and lean body respective muscle mass (measured as urinary creatinine). (**B**) Levels (top) and rate of change (bottom) in urinary testosterone, insulin-like growth factor-binding protein 3 (IGFBP-3) and dehydroepiandrosterone (DHEA) levels.

particularly benefit from parallel measures of IGF-1 as both IGF-1 levels themselves as well as the IGF-1/IGFBP-3 ratio may add to or mediate the effects of IGFBP-3 (*Alberti et al., 2011*). In any case, our results suggest that the GSs found in our study in bonobos mirror human adolescence patterns (*Bogin, 2020*), and therefore, represent adolescent GSs.

Our bonobo results were for the moment limited to zoo populations. Therefore, we cannot rule out that our results may be limited to the zoo-specific environment or genetic variant, and it remains

**Table 2.** Evidence of length growth spurts (GSs) from published literature using linear length growth.

Measures of linear length growth are taken of: Body length or height = B, Crown-rump/Shoulder-rump/Anterior trunk length = CR/SR/AT, Lower/Upper/Full arm length = LA/UA/A, Thigh/Tibia/Leg length = TH/TI/L. Methods: in zoos = direct measurements, in wild populations = photogrammetry, except on *Macaca ochreata* (direct on trapped animals). Growth rate acceleration can be seen as proof of a GS, but considering scale correction, a GS is also very likely in case of a period with constant linear length growth rate, and would be possible in cases of just a slowdown in deceleration. For markers of adolescence see *Appendix 1—table 1*. m = male, *f* = female, w = wild, z = zoo.

| Species (w/z) | Changes in length growth rate | | | | Aligned with weight-GS | Comments | Publication |
|---|---|---|---|---|---|---|---|
| | Acceleration | Constant (plateau) | Slowdown in deceleration | No slowdown in deceleration | | | |
| *Macaca assamensis* (w) | | | m + f (LA) | | Not available | Acceleration if scale corrected | *Anzà et al., 2022*; *Berghänel et al., 2015* |
| *Macaca fuscata* (z) | m + f (B), m (AT, UA) | m (TH, L) | m (LA), f (LA) | f (UA) | Yes (little earlier) | | *Hamada, 1994*; *Hamada et al., 1999*; *Hamada and Yamamoto, 2010* |
| *Macaca nemestrina* (z) | m + f (AT, CR, A, LA, L) | | | | Yes | | *Nishikawa, 1985*; *Tarrant, 1975* |
| *Macaca arctoides* (z) | m (CR) | | | | Yes | Few individuals | *Faucheux et al., 1978* |
| *Macaca mulatta* (z) | (B, TI)[1] | m + f (CR) | | | Yes (little earlier) | [1]Unknown sex, few individuals | *Tanner et al., 1990*[1]; *van Wagenen and Catchpole, 1956* |
| *Macaca ochreata* (w) | m + f (CR) | | | | Yes | | *Schillaci and Stallmann, 2005* |
| *Theropithecus gelada* (w) | m + f (SR) | | | | Not available | | *Lu et al., 2016* |
| *Papio anubis* (z) | m (CRL) | m + f (A) | m (TH), f (CRL, TH) | | Yes (little earlier) | | *Leigh, 2009* |
| *Papio hamadryas* (z) | m (CR) | | | f (CR) | Yes (m) | Coarse data | *Crawford et al., 1997* |
| *Mandrillus sphinx* (z) | | m + f (CR) | | | Yes | | *Setchell et al., 2001* |
| *Pan troglodytes* (z) | | | m + f (B) | | Yes | | *Hamada and Udono, 2002* |
| *Pongo pygmaeus* (z) | m + f (B, LA) | | | | Yes | Two individuals | *Vančatová et al., 1999* |
| *Gorilla beringei beringei* (w) | | | m (B) | m (UA), f (B, UA) | Not available | | *Galbany et al., 2017* |

unknown to which degree they also apply to wild bonobos. However, this affects our main conclusion only marginally, as our results still provide proof that at least some bonobo populations show pronounced adolescent length-GSs, which challenge the hypothesis that an adolescent length-GS is a unique human trait. In addition, our findings have been corroborated by a recent study, which provides initial evidence of an adolescent bone GS in wild-living chimpanzees (*Sandel et al., 2023*).

## Mind the scale

Our findings demonstrate the importance of considering scaling rules between weight and length growth (*Figure 1*). Length-GSs are not reliably detectable or interpretable if only linear length growth is analysed, as they might only appear as a temporary slowdown or plateauing in the continuous decline of linear length growth rate even if they would isometrically align with a pronounced

weight-GS (*Figure 1A–C ,F*; *Cullen et al., 2021*). Previous findings showed similar peak velocities in weight growth in humans compared to non-human primates during the adolescent GS (*Leigh, 2001*; *Leigh, 1996*). Despite this similarity in weight-GS, it has been argued that even if non-human primates showed a length-GS during adolescence, its amplitude would be negligible compared to humans (*Bogin, 2020*). Indeed, the human adolescent length-GS is also evident on the linear scale and more pronounced than in any other primate, including our results on bonobos. However, human body length growth rates during the GS are almost identical with those in chimpanzees, with average peak velocity of height growth in humans being 9–11 cm/year for boys and 7–9 cm/year for girls, and in chimpanzees 8–10 cm/year in males and 6–10 cm/year in females (*Bogin, 2020*; *Hamada and Udono, 2002*). Similarly, also the length growth rates of the forearms in our study match or even exceed human forearm growth rates during the adolescent GS (*Nowak-Szczepanska and Koziel, 2016*). Human-like length growth rates were also found in e.g. captive Japanese macaques, despite their smaller body size (*Hamada et al., 1999*).

However, as outlined in the introduction and *Figure 1*, the occurrence and magnitude of a GS in linear length are largely independent of the magnitude and only depend on the pace and pattern of weight growth acceleration. Theoretically, two human characteristics could contribute to a faster weight growth acceleration, and thus, the remarkable GS in linear length seen in humans. First, humans show a juvenile period of particularly slow weight growth from which the adolescent GS has to rise, which seems absent in non-human primates including our results on bonobos (*Bogin, 2020*; *Leigh, 1996*; *Watts and Gavan, 1982*). Second, the human adolescent GS is squeezed, taking less time relative to body size than in other primates (*Leigh, 2001*). How and to what degree these aspects may influence the pattern of growth rate acceleration during the adolescent GS in humans needs to be addressed in future research. Additionally, the detectability and magnitude of a length-GS increases with the body weight at which the acceleration in weight growth takes off. Here, humans and chimpanzees as well as bonobos have a delayed onset of the adolescent weight-GS compared to other primates (*Leigh, 2001*). Finally, although humans seem to accomplish the same proportion of their adult body weight during the GS as other primates do (*Leigh, 2001*), they accomplish a larger proportion of their adult body length during the GS than at least chimpanzees do (*Smith, 1993*), which could cause a higher detectability on its own and may indicate some differences in allometry between humans and chimpanzees. Therefore, the adolescent length-GS in primates 'would not be an all or nothing thing, but rather a matter of degree' (*Watts and Gavan, 1982*).

Our primate literature search on available length growth data showed that length-GSs might be more widespread and pronounced in primates than previously thought, and might align with weight-GSs and, in some species, with certain indicators of adolescence. Several species showed a small GS in linear length, which sufficiently proves the existence of a length-GS. However, these studies would still benefit from a reanalysis of scale-corrected data, since analyses of linear length growth very likely underestimate the magnitude and age of these length-GSs, and thus their alignment with weight growth. Aligning length-GSs can further be expected in species that show constant growth rates in linear length during their weight-GSs (*Figure 1C*), like in both sexes of mandrils, or in male baboons (*Papio hamadryas*) and female geladas (*Theropithecus gelada*) (*Leigh, 2009*; *Lu et al., 2016*; *Setchell et al., 2001*). These patterns indicate that in non-human primates, weight-GSs and length growth are not decoupled and in 'sharp contrast' (*Bogin, 2020*, p. 180) but rather aligned with each other, and the apparent absence of length-GSs despite in parts pronounced weight-GS is merely a mathematical artefact. Whether these weight- and length-GSs reflect adolescent GSs remains an open question that would require more detailed and parallel measured physiological data on developmental stages at the individual level.

Our results further demonstrate how the consideration of important scaling laws also extends to other arguments. This may for example apply to previous findings that in humans and other primates, length-GSs do not align with but precede weight-GSs (*Hamada et al., 1999*; *Leigh, 2009*; *Tanner et al., 1990*). Again, this could be an artefact of comparing weight and length growth rate at different dimensions. Even if a linear length-GS is detectable, it will by definition reach its peak velocity at an earlier age than the associated weight-GS since linear growth starts to decline already when weight (and exponentiated length) growth rate is still rising (*Figure 1F*), resulting in an artificial time lag even if the weight and length-GSs are perfectly synchronous. In our study, the female peak velocity in linear length growth occurred already 1 year earlier than in the weight-GS and in the scale-corrected

**Table 3.** Sample sizes for measurements of growth (body weight, forearm length, and creatinine) as well as for physiological markers (dehydroepiandrosterone [DHEA], testosterone, and insulin-like growth factor-binding protein 3 [IGFBP-3]).

| Parameter | Number of males/females | Samples per ID (mean ± SD/range/median) | | | Number of zoos |
| --- | --- | --- | --- | --- | --- |
| | | All | Males | Females | |
| Body weight | 119/139 | 32.4 ± 67.6/1–659/9 | 40.4 ± 90.3/1–659/9 | 25.5 ± 37.8/1–195/9 | 19 |
| Arm length | 56/79 | 4.8 ± 3.0/1–11/4 | 4.8 ± 2.9/1–11/5 | 4.8 ± 3.0/1–11/4 | 10 |
| Creatinine | 65/89 | 4.9 ± 3.7/1–19/4 | 5.3 ± 4.1/1–19/4 | 4.7 ± 3.5/1–16/4 | 13 |
| DHEA | 68/87 | 5.1 ± 3.7/1–19/4 | 5.1 ± 4.0/1–19/4 | 5.0 ± 3.5/1–16/4 | 14 |
| Testosterone | 68/89 | 5.1 ± 3.8/1–19/4 | 5.3 ± 4.1/1–19/4 | 5.0 ± 3.5/1–16/4 | 15 |
| IGFBP-3 | 45/61 | 1.5 ± 1.4/1–8/1 | 1.7 ± 1.6/1–7/1 | 1.4 ± 1.2/1–8/1 | 12 |

length-GS, because of the underlying mathematical scaling rules. Hence, the time lag seen in humans and other primates may be a mathematical artefact, altogether.

We showed the importance of keeping scaling relationships in mind when addressing and comparing growth rates. Reporting and comparing them at their different dimensions make proper interpretations very difficult at best, and would (and did) provoke misleading interpretations (*Figure 1*; see also *Cullen et al., 2021*). In our study, we focused on polynomial functions for matter of simplicity (*Figure 1*), but our arguments extend also to other patterns like e.g., an exponential acceleration in growth rate.

Altogether, our results show that if compared at comparable scales, the adolescent GSs in bonobos, and probably primate GSs in general, encompass similar changes in weight and length growth rates which correspond well with each other. Hence, the proposed general uniqueness of the human adolescent GS in length growth can be rebutted.

## Materials and methods
### Study population
All data were collected from zoo-housed bonobos from European and North American zoos. Chronological age of 220 individuals born in zoos was known from zoo records. For the 40 wild-born individuals in our dataset, the average age was estimated as 2.5 years (range 0.1–8 years) when brought into captivity. Our dataset includes long-term measures in body weight and forearm length as well as measures of DHEA, creatinine, specific gravity, IGFBP-3, and testosterone extracted from urine samples. Sample sizes, sex distribution, and underlying numbers of different zoos are provided in *Table 3*.

For all zoo-born individuals, identity of dams and sires was known and taken from the international studbook (*Stevens and Pereboom, 2020*).

### Data collection
#### Body weight data
We collected measures of body weight on 260 individual bonobos housed at 19 different institutions (*Table 3*) from two datasets and three publications. The first and largest dataset consisted of bonobo body weights entered into the Zoological Information Management System (ZIMS) software for Husbandry. ZIMS is a web-based record-keeping system used by zoos, aquariums, and zoological associations to capture and organize husbandry information. We acquired written permission from each of the zoos to use the datasets they entered in ZIMS. Data included zoo records between 1955 and December 2020. The second dataset (703 data points) was compiled at the Royal Zoological Society of Antwerp over the years, by collecting historical data from animals in their collection and written communication with several zoos. From this second dataset, we only used data that were not already in ZIMS. In addition, we used published data from *Hill, 1968* (N = 2), *Jantschke, 1975* (N = 1), and *Neugebauer, 1980* (N = 55). We excluded individuals for which only body weights at death were recorded, as well as data points from pregnant females. For every data point, we entered identity of the individual, sex, date of birth (birthdates and estimates for the wild-born individuals were taken

from the international studbook *Stevens and Pereboom, 2020*), date of weighing, location of weight measurement, and rearing conditions (either wild-born: $N = 40$, zoo-born and hand-reared: $N = 50$, or zoo-born and mother-reared: $N = 170$).

## Forearm length data

Morphometric measurements from bonobos were collected with a transparent Plexiglas tube (125 × 1400 mm, with a metric scale on each side) attached to the enclosures, as previously validated for bonobos (*Behringer et al., 2016a*). Technical information and figures of the device and the procedure are provided in *Behringer et al., 2016a*. Morphometric measurements were taken from digitized images from video recordings (Sony HDR–CX115EB Full HD Camcorder) of individuals inserting their arms into the tube for a reward. The digitized images were analysed using ImageJ (*Abràmoff et al., 2004*). Forearm length was taken from two anatomical surface landmarks, one located distally at the wrist at the depression between the base of the thumb and distal radius, and the second proximally at the point of the posterior depression of the lateral group of forearm extensor muscles and just lateral to the cubital fossa.

## Physiological marker

Urine samples were collected throughout the day between 6:00 and 20:00 hr. Samples were collected on plastic sheets or from the floor with disposable plastic pipettes and transferred into 2 ml plastic vials. Urine was frozen immediately after collection. All samples were transported frozen to the Max Planck Institute for Evolutionary Anthropology in Leipzig, Germany. All measurements were corrected for specific gravity.

## Creatinine

We measured creatinine levels using the Jaffe reaction (*Anestis et al., 2009*; *Jaffe, 1886*).

## DHEA and testosterone

Urinary DHEA and testosterone were measured using liquid chromatography–tandem mass spectrometry. The extraction of testosterone and DHEA from urine was done following the extraction protocol described elsewhere (*Hauser et al., 2008*), with the modifications described in *Wessling et al., 2018*. The extraction method of the urine included a solvolysis, and therefore, the presented urinary measures of DHEA represent a combination of DHEA-S and DHEA concentrations.

## IGFBP-3

The amount of urine per sample was limited, and only when the amount of urine was sufficient after having performed all other physiological measurements, we sent frozen aliquots of the samples to the Laboratory for Translational Hormone Analytics in Paediatric Endocrinology, Center of Child and Adolescent Medicine, Justus-Liebig University, Giessen, Germany for analyses of IGFBP-3. In a pilot study, we were unable to determine IGF-I in bonobo urine samples ($N = 30$ samples). However, urinary IGFBP-3 levels were measured with a radioimmunoassay (RIA) developed for human IGFBP-3 detection (*Blum et al., 1990*) and validated for bonobos (*Behringer et al., 2016b*).

## Specific gravity

We measured specific gravity using a digital hand refractometer (TEC, Ober-Ramstadt, Germany), to correct all urinary physiological measurements for urine concentration (*Miller et al., 2004*).

## Statistical analyses

### General setting for all models

All statistical analyses were performed with R 4.1.3 (*R Development Core Team, 2022*) using packages mgcv 1.8-40 (*Wood, 2017*), itsadug 2.4.1 (*van Rij et al., 2020*) and gratia 0.7.3 (*Simpson, 2020*), and the complete code is provided in the supplemental material. We applied GAMMs (with Gaussian distribution) which allow for the detection and analysis of complex non-linear relationships (termed 'smooths') that are typical for age trajectories like growth curves and changes in physiological markers during development. We used function bam for the body weight models due to large sample size, and

gam for all other models (package mgcv), with smooth estimation based on Maximum Likelihood estimation and penalized cubic regression splines as smooth basis. We checked for model assumptions and appropriate model settings using function gam.check (package mgcv) and acf_resid (detection of autocorrelation, package itsadug), and compared the Full model with the Null model (containing the random effects only) via function compareML (package itsadug). GAMM smooths were plotted using package itsadug (functions plot_data, plot_smooth, and plot_diff, with removed random effects). First-order derivatives of the GAMM results were calculated with function derivatives (package gratia) and plotted with ggplot2 (*Wickham, 2016*). All models had negligible autocorrelation of residuals, apart from the body weight models which showed moderate autocorrelation (rho ≈ 0.3) and were therefore complemented by an AR1 correlation structure term (*Wood, 2017*) which solved the issue.

We ran eight blocks of analyses, investigating sex differences in age trajectories of body weight (in kg and $kg^{1/2.5}$), forearm length (in cm and $cm^{2.5}$), and levels of creatinine, DHEA, testosterone, and IGFBP-3, further controlling for a range of variables (see below). We used the power of 2.5 instead of a cubic relationship (power of 3) as a more conservative approach, following previous results on the scaling relationship between stature height and body weight in bonobos (*Yapuncich et al., 2020*), which we also confirmed for our own dataset (*Figure 2—figure supplement 2*). Data from pregnant females were excluded from all analyses except for forearm length (where exclusion of gestational values also had no effect on the results). We used daily averages in case of multiple body weight measures per day and individual, but refrained from doing so for physiological samples to allow all statistical models to be controlled for daytime of sampling (only relevant for DHEA and creatinine; maximum two samples per day and individual; proportions of samples for which two samples per day were available: DHEA 0.6% of samples and creatinine 0.8% of samples). We log-transformed all physiological response variables (creatinine, DHEA, testosterone, and IGFBP-3) to match model assumptions (Gaussian distribution).

Differences in age trajectories between males and females were investigated by interaction terms between sex and age. As typical for GAMMs (*Wieling, 2018*; *Wood, 2017*), the statistics for these interaction terms were calculated in two ways, first analyzing whether significant changes occur within males and within females, and second whether the smooths of males and females differ significantly from each other (the classic interaction term statistic).

Throughout models, we implemented the same range of random effects. First, we included a random smooth over age per individual and a sex-specific random smooth over age per zoo. First, these random smooth effects controlled for repeated measurements. Moreover, they provide a particular capability of GAMMs to handle the analysis of growth trajectories, that is, to account for individual and zoo-specific variation in non-linear trajectory curves, including random variation in the age of developmental milestones like GS and cease, but also uncertainty in birthdate estimates of wild-born individuals. Therefore, they estimate the global group-specific trajectory curves and their uncertainty behind this variation (*Pedersen et al., 2019*). We further included an interaction term between date of sampling and zoo to control for potential general and zoo-specific changes in bonobo keeping over time such as changes in the composition and caloric content of food, or changes in group composition, enclosure size and breeding management.

Growth trajectories can be influenced by early life conditions particularly during the prenatal and early postnatal period, either directly through programming effects or indirectly through catch-up growth if conditions ameliorate (*Berghänel et al., 2017*; *Lu et al., 2019*). Variation in environmental conditions may arise through maternal effects (maternal age and primiparity) or differences in rearing conditions (hand- vs. mother-reared or wild- vs. zoo-born), and we controlled our analyses for such potential effects. For wild-born individuals, rearing condition and maternal age and parity were unknown. We therefore used a 'composite' categorical variable differentiating between wild- and zoo-born individuals, with zoo-born individuals being further subclassified according to their rearing condition and maternal parity (i.e., from a primi- or multiparous mother and either hand- or mother-reared). Furthermore, we ran another model on zoo-born individuals only, to exclude the unknown variation in parental characteristics and early life conditions in wild-born individuals, and their potential influence on developmental trajectories. Additionally, birthdates were exactly known for all zoo-born but only estimated for wild-born individuals (see above), and forearm data and urine samples of wild-born individuals were only available for bonobos older than 21 and 17 years, respectively. In these reduced models, we could additionally control for effects of maternal age at birth by adding

a non-linear interaction term (tensor product) for the influence of maternal age on age trajectories. Furthermore, we were also able to implement random smooth terms per sire and per dam, thereby controlling for multiple father- and motherhood and heritable parental effects. However, in all models other than for body weight, if random smooth terms per sire and per dam were also added, we could only implement random intercepts per individual due to sample size constraints.

Because of the different sources and sampling periods for body weight, forearm length data, and urine samples, as well as the limited availability of urinary IGFBP-3 levels, there were some individuals for which not all measures were available. Therefore, we re-run our analyses on our main variables (body weight, forearm length, DHEA, and testosterone) on a subsample of individuals (*N* = 44 males and 64 females) for which data on all these four variables was available, which yielded identical patterns (*Figure 2—figure supplement 1*, *Figure 3—figure supplements 1 and 2*).

The model results were further used to calculate the first-order derivatives of the age trajectories, representing the rate of change of the response variable over time (e.g., for body weight the growth rate in kg/year).

### Specific model settings for testosterone and IGFBP-3 levels

Previous studies on age trajectories of testosterone in bonobos showed a strong and rather fast and sudden rise in both males and females (*Behringer et al., 2014*), which was also evident in the raw data of our study but could not be modelled with automatic estimation of smoothing parameters as this resulted in oversmoothing (*Figure 3—figure supplement 3*). Therefore, we fixed the smoothing parameter to sp = 1 in all testosterone models, which allowed for higher wiggliness of the age trajectory of testosterone levels and solved the issue.

The IGFBP models deviated from the specifications described above due to sample size constraints. The composite variable of wild-born, maternal parity and rearing condition was not included. We did not control for date of sampling but for random smooth effects over age per zoo. Furthermore, the individuals' respective sire and dam were included as random intercept instead of random smooth because 83% of the individuals were only sampled once (maximum: nine samples).

## Acknowledgements

The authors thank the directors, curators, and keepers of the zoos of Columbus Zoo and Aquarium; Cologne Zoo; Leipzig Zoo; Milwaukee County Zoo; Vallée des Singes Romagne; Wilhelma Zoo Stuttgart; Zoo Berlin; Zoo Frankfurt, and Zoo Planckendael. Special thanks go to the caregivers of these facilities, and to Marjolein Osieck and Jonas Verspeek for their assistance with forearm measures and urine collection. In addition, we thank Antwerp Zoo, Apenheul Primate Park, Apeldoorn, Cincinnati Zoo and Botanical Gardens, Jacksonville Zoo, Fort Worth Zoo, Memphis Zoo, San Diego Zoo, San Diego Safari, Twycross Zoo, Wuppertal Zoo, as well as the zoos mentioned above for granting access to their records of bonobo body weights to be included in this study. We are grateful to Ròisìn Murtagh, Vera Schmeling, and Meike Schäfer for help in the lab. We would like to thank Stefan A Wudy and Werner F Blum for their support with analyzing samples at the Justus-Liebig University, Giessen, Germany. Special thanks go to Oliver Schülke (University of Göttingen, Germany) for help and useful comments on an earlier draft of this manuscript. Funding: Deutsche Forschungsgemeinschaft (DFG) grant BE 5511/4-1 (VB). Funding for laboratory analyses was provided by the Max Planck Society.

## Additional information

### Funding

| Funder | Grant reference number | Author |
| --- | --- | --- |
| Deutsche Forschungsgemeinschaft | BE 5511/4-1 | Verena Behringer |

The funders had no role in study design, data collection, and interpretation, or the decision to submit the work for publication.

## Author contributions

Andreas Berghaenel, Conceptualization, Software, Formal analysis, Validation, Visualization, Methodology, Writing – original draft, Writing – review and editing; Jeroen MG Stevens, Conceptualization, Resources, Data curation, Funding acquisition, Investigation, Writing – original draft, Writing – review and editing; Gottfried Hohmann, Conceptualization, Funding acquisition, Investigation, Writing – original draft, Writing – review and editing; Tobias Deschner, Resources, Funding acquisition, Validation, Methodology, Writing – review and editing; Verena Behringer, Conceptualization, Data curation, Funding acquisition, Validation, Investigation, Methodology, Writing – original draft, Project administration, Writing – review and editing

## Author ORCIDs

Andreas Berghaenel ⓘ http://orcid.org/0000-0002-3317-3392
Jeroen MG Stevens ⓘ http://orcid.org/0000-0001-6730-4824
Verena Behringer ⓘ http://orcid.org/0000-0001-6338-7298

## Ethics

Forearm measurements: Apes inserted their arms spontaneously into the measuring device. No training, like positive reinforcement training, was necessary. Urine collection was non-invasive and did not harm or disturb the bonobos. Data collection followed the Animal Behavior Society's guidelines for the treatment of animals in behavioral research and teaching and adhered to the standards as defined by the European Union Council Directive 2010/63/EU on the protection of animals used for scientific purposes.

Reviewer #1 (Public Review): https://doi.org/10.7554/eLife.86635.3.sa1
Reviewer #2 (Public Review): https://doi.org/10.7554/eLife.86635.3.sa2
Author Response https://doi.org/10.7554/eLife.86635.3.sa3

## Data availability

Source data and R-code for statistics and figures in the paper are permanently stored at Phaidra (https://doi.org/10.34876/j151-bf68 and https://doi.org/10.34876/ajqf-js44).

The following datasets were generated:

| Author(s) | Year | Dataset title | Dataset URL | Database and Identifier |
| --- | --- | --- | --- | --- |
| Berghänel A, Stevens JMG, Hohmann G, Deschner T, Behringer V | 2023 | Data set for the paper "Adolescent length growth spurts in bonobos and other primates - Mind the scale" | https://doi.org/10.34876/ajqf-js44 | Phaidra, 10.34876/ajqf-js44 |
| Berghänel A, Stevens JMG, Hohmann G, Deschner T, Behringer V | 2023 | R-code for the paper "Adolescent length growth spurts in bonobos and other primates - Mind the scale" | https://doi.org/10.34876/j151-bf68 | Phaidra, 10.34876/j151-bf68 |

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

# Appendix 1

**Appendix 1—table 1.** Evidence of (adolescent) length growth spurts (GSs) from published literature using linear length growth. Same table as *Table 2* in main text, but additionally with literature on markers of adolescence. Changes in length growth rate (left) – Measures of linear length growth are taken of: Body length or height = B, Crown-rump/Shoulder-rump/Anterior trunk length = CR/SR/AT, Lower/Upper/Full arm length = LA/UA/A, Thigh/Tibia/Leg length = TH/TI/L. Methods: in zoos = direct measurements, in wild populations = photogrammetry, except on Macaca ochreata (direct on trapped animals). Growth rate acceleration can be seen as proof of a GS, but taking into account scale correction, a GS is also very likely in case of a period with constant linear length growth rate, and would be possible in cases of just a slowdown in deceleration. Markers of adolescence (right, often different study population): Testes size growth = TS, Rise in testosterone levels = TL, Menarche = M, First swelling/cycle/ovulation = S/C/O; Timing compared to GS: aligned = a, preceding = p, later (following) = l; m = male, f = female, w = wild, z = zoo.

| Species (w/z) | Changes in length growth rate | | | | | | | Markers of adolescence | | |
| --- | --- | --- | --- | --- | --- | --- | --- | --- | --- | --- |
| | Acceleration | Constant (plateau) | Slowdown in deceleration | No slowdown in deceleration | Aligned with weight-GS | Comments | Publication | Males | Females | Publication |
| *Macaca assamensis* (w) | | | m + f (LA) | | Not available | Acceleration if scale corrected | *Anzà et al., 2022; Berghänel et al., 2015* | / | / | / |
| *Macaca fuscata* (z) | m + f (B), m (AT, UA) | m (TH, L) | m (LA), f (LA) | f (UA) | Yes (little earlier) | | *Hamada, 1994; Hamada et al., 1999; Hamada and Yamamoto, 2010* | TS (l) | M (a) | *Hamada et al., 1999* |
| *Macaca nemestrina* (z) | m + f (AT, CR, A, LA, L) | | | | Yes | | *Nishikawa, 1985; Tarrant, 1975* | / | S (p) | *Muehlenbein et al., 2005; Hadidian and Bernstein, 1979* |
| *Macaca arctoides* (z) | m (CR) | | | | Yes | Few individuals | *Faucheux et al., 1978* | TS (a) | / | *Nieuwenhuijsen et al., 1987* |
| *Macaca mulatta* (z) | (B, TI)[1] | m + f (CR) | | | Yes (little earlier) | [1]Unknown sex, few individuals | *Tanner et al., 1990; van Wagenen and Catchpole, 1956* | / | S (a), M (l) | *Tanner et al., 1990* |
| *Macaca ochreata* (w) | m + f (CR) | | | | Yes | | *Schillaci and Stallmann, 2005* | / | / | / |
| *Theropithecus gelada* (w) | m + f (SR) | | | | Not available | | *Lu et al., 2016* | TL (l) | Pigmentation of bare area (a) | *Beehner et al., 2009; Matthews, 1956* |
| *Papio anubis* (z) | m (CRL) | m + f (A) | m (TH), f (CRL, TH) | | Yes (little earlier) | | *Leigh, 2009* | TS, TL, IGF1, IGFBP3 (all a) | C (a) | *Bernstein et al., 2013; Bernstein et al., 2008; Mueller, 2005; Owens, 1976* |
| *Papio hamadryas* (z) | m (CR) | | | f (CR) | Yes (m) | Coarse data | *Crawford et al., 1997* | TS, TL, IGF1, IGFBP3 (all a) | C (a) | *Bernstein et al., 2013; Mueller, 2005* |
| *Mandrillus sphinx* (z) | | m + f (CR) | | | Yes | | *Setchell et al., 2001* | TS (a), TL (l) | S (a) | *Setchell and Dixson, 2002; Wickings and Dixson, 1992* |
| *Pan troglodytes* (z) | | | m + f (B) | | Yes | | *Hamada and Udono, 2002* | TS and TL (a) | M (l) | *Anestis, 2006; Coe et al., 1979; Kraemer et al., 1982* |
| *Pongo pygmaeus* (z) | m + f (B, LA) | | | | Yes | Two individuals | *Vančatová et al., 1999* | Highly variable | M (occurs at 5–12 yrs) | *Nacey Maggioncalda and Sapolsky, 2002; Markham, 1990* |
| *Gorilla beringei beringei* (w) | | m (B) | | m (UA), f (B, UA) | Not available | | *Galbany et al., 2017* | / | / | / |

