## [Editor Report · eLife assessment]

This **valuable** paper sheds new light on the growth trajectory of bonobos (*Pan paniscus*), with explicit contributions to discussions of the exclusivity of certain aspects of growth in modern humans, most specifically with respect to components of the adolescent growth spurt, which may be less human-specific among primates than presumed to this point. The results are **solid**, based on the largest sample ever considered in the study of bonobo growth and include both morphometric and endocrinological data. This work will be of interest to human evolutionary biologists, primatologists, and researchers studying the ontogeny and evolution of growth and development in general.

---

## [Referee Report · Reviewer #1 (Public Review)]

The manuscript provides analyses on a very complete dataset on weight and length growth, as well as several physiological markers related to growth, in bonobos. Moreover, there is a good overview of the presence of adolescent growth spurts in non-human primates, by reviewing published data, in comparison to their own dataset. They discuss the need to consider scaling laws when interpreting and comparing growth curves of different species and variables.

The manuscript is very well written, the sample is large, and the methods are well explained. It seems they have analyzed a very complete dataset. Also, the discussion and the references supporting the findings are complete.

The main weakness of this manuscript is that they do not provide a direct comparison with previously analyzed datasets in other species, using their own method (in part maybe because there is not available data, but just published figures).

On the other side, conclusions are well supported by the results, and the previously published datasets are discussed in the manuscript, although not in detail.

---

## [Referee Report · Reviewer #2 (Public Review)]

This work sheds new light on the growth trajectory of Bonobo and contributes heavily to the discussion of the exclusivity of certain aspects of growth in modern humans. These results are also interesting as long as they are based on the study of the largest sample ever considered in the study of the growth of this species by including morphometric measurements as well as endocrinological factors.

The authors approach the study of the presence of growth spurs (GS) in Bonobo on the basis that GS are exclusive to the growth in modern humans. This idea is fairly widespread, however studies on non-human primates have shown an acceleration of growth during adolescence in several species, these works are recalled, presented and discussed by the authors. The originality of this work lies in highlighting the importance of scaling in studies of growth trajectories. The absence of GS in Bonobo but also in other primate species may result from not considering the conjunction of weight and height in the analysis of growth, because the pronounced changes in the speed of the height are in relation to the speed of changes in weight and this is modified according to the size/age. The authors apply scaling corrections to their results and the GS become evident (or more obvious) in Bonobo. Thus, the exclusivity of GS in growth in modern humans may in fact result only by the application of analytical approach not very appropriate in non-human primates.

---

## [Author Response]

The following is the authors’ response to the original reviews.

**Reviewer #1 (Recommendations For The Authors):**
I would recommend the authors check the results section, it seems to me that the first two paragraphs are not results, but methods.

We would like to express our appreciation to both reviewers for bringing this to our attention. Indeed, we discussed this in detail, but decided that because the methods come after the results section. We believe that providing the basic methodological approach to readers before the results is essential for better comprehension. Once again, we sincerely thank the reviewers for their valuable feedback, however, we would prefer to leave this part as it is.

In Figure 3B, why there is not male and female shown in different lines, as in the rest of figures? I recommend following the same pattern everywhere.

Has been changed accordingly, and the respective sex-specific lines were also added to Figure 4.

I recommend checking carefully all the articles included in Table 2. Maybe some of the included information here is not precise.

We thank the reviewer for highlighting this. We carefully checked the articles again, and made some small adjustments.

In Material and methods: just note that when ages are estimated, usually there is a variable accounting for the amount of estimated years, that should be included in the model, and see that it has no effect on the dependent variable. I recommend including this variable.

We sincerely appreciate the helpful comment from the reviewer, which we have carefully considered and implemented in our manuscript. However, we would like to highlight that addressing age estimation error is complex, as it involves measurement error. Thus, simply adding it as an independent variable may not fully capture its potential impact, as the effect may be positive or negative depending on the individual. Hence, the potential effect would be better accounted for by the implementation of individual random intercepts and smooths to adjust the confidence intervals, which is part of our model structures. Furthermore, we would like to emphasize that we have also conducted analyses on a reduced dataset that only included zoo-born individuals with precisely known birthdates, and the results remained consistent. So instead of changing our analyses, we now emphasize how our approach also addresses this aspect.

Creatinine: Is there any other reference, more recent and in English, to complement the original one cited?

We have now supplemented the original citation with an additional English citation: Anestis et al. 2009.

**Reviewer #2 (Recommendations For The Authors):**
Minor correctionsPlease, in Study population, the citation of table 2 is in fact Table 3. For table 3 (in Methodology), please provide the units Body weight having a mean of 32.4, has it a median of 9 ?Please, provide results separately for males and females

We changed the table as requested, though the table only reports sample sizes and thus only numbers without units. The values for body weight are accurate.

In ResultsThe two first paragraphs have to be included in methods and structured with those already present.

We would like to express our appreciation to both reviewers for bringing this to our attention. Indeed, we discussed this in detail, but decided that because the methods come after the results section, we believe that providing the basic methodological approach to readers before the results is essential for better comprehension. Once again, we sincerely thank the reviewers for their valuable feedback, however, we would prefer to leave this part as it is.

In Table 1, indicate what 'Est' means.

Has been changed accordingly